# An Optimization Method of Precision Assembly Process Based on the Relative Entropy Evaluation of the Stress Distribution

**DOI:** 10.3390/e22020137

**Published:** 2020-01-23

**Authors:** Zifu Wang, Zhijing Zhang, Xiao Chen, Xin Jin

**Affiliations:** School of Mechanical Engineering, Beijing Institute of Technology, Beijing 100081, China; wangzifubit@163.com (Z.W.); chenxiao_bit@163.com (X.C.)

**Keywords:** relative entropy, assembly process optimization, assembly stress, stress distribution, strain energy density

## Abstract

The entropy evaluation method of assembly stress has become a hot topic in recent years. However, the current research can only evaluate the maximum stress magnitude and stress magnitude uniformity, and it cannot evaluate the stress position distribution. In this paper, an evaluation method of stress distribution characterized by strain energy density distribution is proposed. In this method, the relative entropy is used as the evaluation index of the stress distribution difference between the error model and the ideal model. It can evaluate not only the stress magnitude, but also the stress position. On this basis, an optimization method of the precise assembly process which takes the relative entropy as the optimization objective is proposed. The stress distributions of the optical lens are evaluated, and the assembly angle of the spacer in the process of the optical lens system assembly is optimized. By comparing the stress distribution of the optimized model and the ideal model, the validity of this method is proved.

## 1. Introduction

In the production process of parts, geometric errors are produced due to the influence of systematic errors and uncertain factors [1]. On the micro scale, the geometric error is represented by the peaks and valleys of the part’s surface, which cause the positioning deviation and contact stress [2]. The geometrical error will lead to non-uniform stress distribution, and then to the non-uniform deformation of parts. Finally, the precision drift of the precision mechanical structure occurs with the long-term storage and temperature change. Improving the machining accuracy of parts can improve this situation, but it also means high cost. It is necessary to optimize the assembly process in order to minimize the stress nonuniformity caused by geometric errors. In this way, the precision drift of the assembly system is reduced and the machining accuracy of parts does not need to be improved. At present, the research of assembly process optimization mainly focuses on improving assembly efficiency [3], reducing cost [4], and the methods they used are mainly tolerance optimization based on statistical deviation [5]. Meanwhile, the studies on the evaluation of stress distribution are mainly qualitative under the hypothetical deviation, which cannot be the indices needed for optimization. That is to say, there is no existing assembly process optimization method that can effectively optimize the stress distribution of parts with real geometric errors.

The evaluation of stress distribution is directly related to the construction of optimization objectives. Evaluation methods of stress distribution are widely used in engineering [6,7] and medicine [8,9]. The evaluation of stress distribution includes the evaluation of the uniformity of stress magnitude and the evaluation of stress position distribution. Since information entropy represents the uncertainty of random events and it is widely used to represent the uniformity of data, some researchers used information entropy to evaluate stress magnitude uniformity. Bonakdari et al. [10] studied the application of Tsallis entropy in the prediction of shear stress distribution in open channel. Khozani [11] used the maximum entropy theory to maximize the Renyi entropy under certain constraints and deduced the shear stress distribution. Zhou Shujing et al. [12] studied the optimization design method of the section size of truss structure. The information entropy is used to represent the stress uniformity of truss structure. When the stress of each member is equal, the safety of the structure is the best. Zhang Xiaojun et al. [13] studied the evolution of rock system and characterized the ordered degree of rock stress field with entropy. When the rock was gradually destroyed, the entropy decreased. Yan Fang et al. [14] proposed an entropy-based evaluation method of contact strain energy to predict assembly accuracy. Xiao Chen et al. [15] proposed a more effective method to predict the stability of assembly accuracy by replacing the entropy of strain energy density with the entropy of contact pressure.

However, researchers mainly focus on the uniformity evaluation of stress magnitude (or strain energy density) using entropy, but cannot consider the distribution of stress position at the same time. If only the uniformity of the stress magnitude is considered, the entropy results are equal in some cases, but the stress position distributions of them are completely different. Furthermore, in complex mechanical structures, the optimal stress distribution caused by assembly force is often coupled of uniformity and non-uniformity, which cannot be directly evaluated by entropy. Therefore, the existing research cannot fully and effectively evaluate the assembly stress distribution in complex cases.

In view of the above challenges, this paper proposes a relative entropy evaluation method to evaluate the difference between the stress distribution of the error model and the ideal model. Relative entropy is widely used in many fields [16,17]. It is an asymmetric measure of the difference between two probability distributions [18,19]. When two probability distributions are the same, their relative entropy is 0. When the difference between two probability distributions increases, their relative entropy will also increase. As is known to all, the stress distribution of the ideal model is the best possible state of the parts under the existence of assembly force, and it is also the best and most stable state of assembly system quality. The relative entropy can reflect the difference between the stress distribution of the error model and the best stress distribution of the idea model, as the evaluation of the stress distribution. After that, assembly process parameters, such as assembly angle or assembly force, can be optimized to obtain the assembly system with the best stress distribution.

The stress distribution evaluation method and the corresponding optimization method proposed in this paper are based on the geometric error modeling. In terms of geometrical error modeling of parts, most studies only consider the statistical tolerance and deviation [20,21,22,23,24], which cannot reflect the real geometric errors of parts. Zhongqing Zhang [2] presents a modeling method of a solid model with real geometric error surfaces in CAD, as shown in Figure 1. In this paper, on the basis of this modeling method, the solid model with real geometric error surfaces of the parts is built, assembled, and imported into the finite element software.

For clarity, the three-dimensional assembly finite element model with real geometric error surfaces is hereinafter referred to as the error model. Accordingly, the three-dimensional assembly finite element model with ideal surfaces is called the ideal model.

In this paper, an optical lens system is taken as the research object. An optical lens system is an essential part of machine vision systems, which is mainly composed of a lens barrel, spacers, lenses, and a threaded ring. A typical optical lens system structure is shown in Figure 2, and the parts’ functions are shown in Table 1. The magnitude and distribution of stress are important performance indexes of a lens. The existence of stress seriously damages the imaging quality of optical lens elements. In the case of machining, heat treatment, and improper installation during production, the internal stress of lens will occur [25]. The non-uniform stress will reduce the optical uniformity of the lens, resulting in the inconsistent distribution of refractivity [26]. In serious cases, it may cause optical failure due to poor imaging quality. Annealing stress also has an important influence on the imaging quality of optical system [27].

In addition to the stress introduced in the manufacturing process of the lens itself, there is also the stress generated by the assembly force in the assembly process, including the preload brought by the threaded ring and the tension brought by the adhesive bonding. Ideally, the preload of the threaded ring should be uniformly applied to the outermost ring area of the lens. However, due to the geometric errors of threaded ring and spacer, the contact is not uniform, which also results in stress concentration.

In order to explain the difference of stress distribution between error model and ideal model, the contact between a spacer and an optical lens in the optical lens system is taken as an example. The lens is an ideal model. In order to illustrate the stress distribution characteristics of the lens, the von Mises stress nephogram of the lens surface calculated with finite element software is shown in Figure 3, and the setting of finite element parameters is shown in Section 3. If the spacer is an ideal part, the contact with the lens will be uniform. The stress is evenly distributed in the circular direction of the lens, and in the radial direction of the lens, the stress decreases from the direct contact ring area to the center of circle, as shown in Figure 3a. If there is a geometric error surface on the spacer, the lens will not be able to contact evenly and will result in local stress concentration, as shown in Figure 3b. The stress change curve of lens radial direction of the ideal model and the error model is shown in Figure 4. The stress distribution curve of the ideal model lens in any radius direction is almost the same; the error model lens has different stress distribution curves in different radius directions. The stress of error model is greater than that of ideal model as a whole.

The stress distribution of the ideal model is the best possible state of the parts under the existence of assembly force, and it is also the best and most stable state of optical lens system imaging quality. Generally, the lens is assembled between the upper and lower spacers. The upper and lower spacers have their own geometric errors. By changing the assembly angle of the spacer, the stress distribution of the lens can be changed. By optimizing the assembly angle of the spacer, the stress distribution of the lens can approach the ideal model to the greatest extent, which means that the stress distribution of the lens is the best.

This paper presents a stress distribution evaluation method based on relative entropy. The assembly process of precision optical lens system is optimized. This method is universal, which can evaluate the stress distribution of parts quickly and effectively, and then the assembly process can be optimized under the guidance of the method.

## 2. Methodology

In information theory, if there is a discrete random variable *X*, for *x*∈*X*, its occurrence probability distribution is *P(x)*, then the entropy *H* of *X* is defined as:(1)$$H\left(X\right)=-\underset{x\in X}{\sum }P\left(x\right)logP\left(x\right)$$

In Equation (1), the more uniform the probability distribution *P(x)*, the greater the entropy *H(x)*. When all *x*∈*X* are equal, which means their occurrence probabilities are equal, the entropy *H(x)* is at its largest. Therefore, entropy can represent the uniformity of the occurrence probability distribution of a random variable.

Assuming that there is a theoretical probability distribution *Q(x)* for *x*∈*X*, the relative entropy can measure the difference between the theoretical probability distribution and the actual probability distribution. The calculation formula of relative entropy [18] is:(2)$$H_{R}\left(X\right)=\underset{x\in X}{\sum }\left[P\left(x\right)logP\left(x\right)-P\left(x\right)logQ\left(x\right)\right]=\underset{x\in X}{\sum }P\left(x\right)log\frac{P\left(x\right)}{Q\left(x\right)}$$
where $H_{R}\left(X\right)$ is the relative entropy, *P(x)* is the real probability distribution, and *Q(x)* is the theoretical probability distribution. The closer *P(x)* and *Q(x)* are, the smaller the relative entropy $H_{R}\left(X\right)$ is. If the two probability distributions of *P(x)* and *Q(x)* are identical, the relative entropy $H_{R}\left(X\right)$ is equal to 0.

A hypothetical ideal case and two error cases are shown in Figure 5. The entropy results calculated in the two error cases of Figure 5 are equal, as shown in Table 2. However, the position distributions of the points in two error cases are totally different.

The hypothetical ideal case is taken as the theoretical probability distribution *Q(x).* The relative entropy evaluation results of two error cases are different. Relative entropy can distinguish the position distribution of the points. The entropy and relative entropy evaluation results of the two error cases are shown in Table 2. The relative entropy of error case 2 is smaller than that of error case 1, which means the point position distribution of error case 2 is more similar with the hypothetical ideal case than error case 1.

The evaluation method in this paper is to calculate the strain energy density instead of the stress. Because the characteristic parameters of the stress field in the entropy model need to be in the form of scalar, the stress vector cannot be directly used for entropy evaluation. When an object is subjected to external forces, the stress and strain will be stored in the form of potential energy which is called strain energy. Converting strain energy to strain energy density can eliminate the volume effect caused by different element division methods. In this way, the stress field can be expressed as a scalar form [14].

For simplicity, strain energy density is hereinafter referred as SED. The SED of the element can be directly output by using the finite element software.

The probability distribution of SED $P^{E}$ of the error model is taken as the true probability distribution *P(x)*; the probability distribution $P^{I}$ of the ideal model is taken as the theoretical probability distribution *Q(x)*. The more similar $P^{E}$ is to $P^{I}$, the closer their relative entropy is to 0. The SED distribution of the ideal model is the theoretical optimal solution in the presence of assembly force. Therefore, the more similar $P^{E}$ and $P^{I}$, the better the SED distribution of the error model. The flow chart of the relative entropy evaluation method of SED distribution is shown in Figure 6.

### 2.1. Construction of the SED Probability Distribution

The key of using relative entropy to evaluate stress distribution is to build a proper stress probability distribution.

Firstly, the SED of each element in the finite element calculation is derived, and the SED concentration element is selected. The elements whose SED are larger than the median are selected as the SED concentration elements. The median can ensure that the number of SED concentration elements selected in each calculation of the same model is equal, and avoid the impact of changes in the number of elements.

Secondly, SED concentration elements are grouped according to magnitude. It is divided into *M* groups from the minimum value to the maximum value, and the number of elements in each group is calculated as follows:(3)$$N_{L}=\left[N_{1},N_{2},\ldots ,N_{m},\ldots ,N_{M}\right]\;\;\left(m=1,2,\ldots ,M\right)$$
where $N_{L}$ is the number of elements in each group, and *M* is the total number of groups.

Thirdly, the SED concentration elements which have been divided into *M* groups are grouped again into *P*Q* groups according to their positions. The lens is divided evenly into *P* sector regions along the circular direction, and then into *Q* ring regions along the radius direction. There are *P*Q* regions in total. The schematic diagram of lens region division is shown in Figure 7.

The number of SED concentration elements in each group is calculated as follows:(4)$$N_{lc}=\left[\begin{array}{c} N_{111},N_{112},\ldots ,N_{1pq},\ldots ,N_{1PQ}\\ N_{211},N_{212},\ldots ,N_{2pq},\ldots ,N_{2PQ}\\ \begin{array}{c} \ldots \\ N_{n11},N_{n12},\ldots ,N_{mpq},\ldots ,N_{MPQ} \end{array} \end{array}\right]\;\left(\begin{array}{ccc} m=1,2,\ldots ,M & p=1,2,\ldots ,P & q=1,2,\ldots ,Q \end{array}\right)$$
where $N_{lc}$ is the number of elements in each group after the grouping according to the SED concentration elements magnitude and position. *M*, *P*, and *Q* are the number of groups respectively.

Finally, Equation (4) is divided by the total number of SED concentration elements. The probability distribution of SED concentration elements considering the magnitude and position *P* is obtained:(5)$$P=\frac{N_{lc}}{\sum N_{mpq}}=\left[\begin{array}{c} P_{111},P_{112},\ldots ,P_{1pq},\ldots ,P_{1PQ}\;\\ P_{211},P_{212},\ldots ,P_{2pq},\ldots ,P_{2PQ}\\ \begin{array}{c} \ldots \\ P_{n11},P_{n12},\ldots ,P_{mpq},\ldots ,P_{MPQ} \end{array} \end{array}\right]\;\left(\begin{array}{ccc} m=1,2,\ldots ,M & p=1,2,\ldots ,P & q=1,2,\ldots ,Q \end{array}\right)$$

### 2.2. The Relative Entropy Evaluation Method of SED Distribution

The finite element software is used to simulate the ideal model and the error model, and the element coordinates and element SED of the target parts in the simulation results are derived. Then, the SED concentration elements are selected and grouped. The probability distributions of the ideal model and the error model are constructed:(6)$$P^{I}=\frac{{N_{lc}}^{I}}{\sum {N_{mpq}}^{I}}=\left[\begin{array}{c} P_{111}{}^{I},P_{112}{}^{I},\ldots ,P_{1pq}{}^{I},\ldots ,P_{1PQ}{}^{I}\\ P_{211}{}^{I},P_{212}{}^{I},\ldots ,P_{2pq}{}^{I},\ldots ,P_{2PQ}{}^{I}\\ \begin{array}{c} \ldots \\ P_{n11}{}^{I},P_{n12}{}^{I},\ldots ,P_{mpq}{}^{I},\ldots ,P_{MPQ}{}^{I} \end{array} \end{array}\right]$$
(7)$$P^{E}=\frac{N_{lc}{}^{E}}{\sum N_{mpq}{}^{E}}=\left[\begin{array}{c} P_{111}{}^{E},P_{112}{}^{E},\ldots ,P_{1pq}{}^{E},\ldots ,P_{1PQ}{}^{E}\\ P_{211}{}^{E},P_{212}{}^{E},\ldots ,P_{2pq}{}^{E},\ldots ,P_{2PQ}{}^{E}\\ \begin{array}{c} \ldots \\ P_{n11}{}^{E},P_{n12}{}^{E},\ldots ,P_{mpq}{}^{E},\ldots ,P_{MPQ}{}^{E} \end{array} \end{array}\right]$$
$$\left(\begin{array}{ccc} m=1,2,\ldots ,M & p=1,2,\ldots ,P & q=1,2,\ldots ,Q \end{array}\right)$$
where $P^{I}$ and $P^{E}$ are the ideal model and the error model probabilities distributions of SED concentration elements respectively. The relative entropy $H_{R}$ can be obtained by taking $P^{I}$ and $P^{E}$ into Equation (2):(8)$$H_{R}\left(X\right)=\underset{x\in X}{\sum }P^{I}\left(x\right)log\frac{P^{I}\left(x\right)}{P^{E}\left(x\right)}$$
where $H_{R}$ is the relative entropy. $P^{I}\left(x\right)$ represents the true distribution of data, that is, the probability distribution of error model SED concentration elements. $P^{E}\left(x\right)$ represents the theoretical distribution of data, that is, the probability distribution of ideal model SED concentration elements. The smaller the relative entropy, the smaller the difference between the error model and the ideal model, the better the error model SED distribution.

### 2.3. Case Study

A few simple cases are used to explain the relative entropy evaluation method of SED distribution. Elements of the model are replaced by points in cases. X and Y coordinates represent the positions of elements, and Z coordinates represent the SED of the elements. The position and SED are settled according to the stress change pattern of the ideal model and the error model in Figure 4b. The Z coordinate is displayed by the color bar, as shown in Figure 8. Figure 8a is the ideal case, and Figure 8b is an error case 1 which Z coordinate (SED) is random. It should be emphasized that the random cases cannot be completely equivalent to the results of finite element calculation. Since the stress and SED between adjacent elements are interrelated in the finite element calculation results, there is no such correlation in the random points. Therefore, this case only concerns the similarity between the error model and the ideal model and the verification of the relative entropy calculation method.

#### 2.3.1. Determining the Value of Divided Regions

In Section 2.1, the values of *M*, *P*, and *Q* of the divided regions obviously affect the results of relative entropy evaluation. *M*, *P*, and *Q* should match the total number of elements in the model. Too few regions will lead to inaccurate evaluation results; too many regions will increase the amount of calculation, without improving the accuracy of calculation. The purpose of determining the values of *M*, *P*, and *Q* is to minimize their impact on the evaluation results.

In order to determine the best values of divided regions, *M*, *P*, and *Q* in error case 1 are increased, and the relative entropy is calculated in turn. First, the value of *M* is increased, the initial value of *P* (*P* = 20) is kept unchanged, and then *Q* is increased from 1 to 5 for calculation, as shown in Figure 9.

It can be seen from Figure 9 that with the increasing of *M* and *Q*, the relative entropy gradually increases and reaches stability after *M* > 10 and *Q* > 2. In order to minimize the impact on the evaluation results, *M* = 11 is taken at the place where the curve goes smoothly.

Next, *M* is kept *M* = 11 unchanged, the value of *P* is increased, and then *Q* is increased from 1 to 5 for calculation. The results are shown in Figure 10.

In Figure 10, with the increasing of *P* and *Q*, the relative entropy decreases gradually and reaches stability after *P* > 35 and *Q* > 2, and *P* = 40 is taken at the stable position of curve change.

Finally, keep *M* = 11, *P* = 40, and *Q* value is increased in turn for calculation. The results are shown in Figure 11.

In Figure 11, with the increasing of *Q*, the relative entropy increases gradually and reaches stability after *Q* > 10, and *Q* = 12 is taken at the stable position of curve change. Thus far, the values of *M*, *P*, and *Q* are completed:M=11 P=40 Q=12

For the same type of model, the number and distribution of SED concentration elements are similar. Therefore, by determining the *M*, *P*, and *Q* of one case, all similar cases can be calculated. If the element number of the model increases or the model shape changes, *M*, *P*, and *Q* are supposed to be determined again.

#### 2.3.2. The Relative Entropy Evaluation

Four cases were randomly generated as error models: error case 1, error case 2, error case 3, and error case 4. The elements SED and position are shown in Figure 12.

Firstly, the point whose SED is greater than the median is selected as the SED concentration points. Points are grouped according to the *M*, *P*, and *Q* values determined in Section 2.3.1. According to Equation (4), the number of SED concentration points in each region is calculated in turn. Then, according to Equations (6) and (7), the SED probability distributions of ideal case and each error case are constructed:(9)$$P^{I}{}_{1}=\frac{N_{lc}{}^{I}}{\sum N_{mpq}{}^{I}}=\left[\begin{array}{c} P_{111}{}^{I},P_{112}{}^{I},\ldots ,P_{1pq}{}^{I},\ldots ,P_{1\;40\;12}{}^{I}\\ P_{211}{}^{I},P_{212}{}^{I},\ldots ,P_{2pq}{}^{I},\ldots ,P_{2\;40\;12}{}^{I}\\ \begin{array}{c} \ldots \\ P_{n11}{}^{I},P_{n12}{}^{I},\ldots ,P_{mpq}{}^{I},\ldots ,P_{11\;40\;12}{}^{I} \end{array} \end{array}\right]$$
(10)$$P^{E}{}_{1}=\frac{N_{lc}{}^{E}}{\sum N_{mpq}{}^{E}}=\left[\begin{array}{c} P_{111}{}^{E},P_{112}{}^{E},\ldots ,P_{1pq}{}^{E},\ldots ,P_{1\;40\;12}{}^{E}\\ P_{211}{}^{E},P_{212}{}^{E},\ldots ,P_{2pq}{}^{E},\ldots ,P_{2\;40\;12}{}^{E}\\ \begin{array}{c} \ldots \\ P_{n11}{}^{E},P_{n12}{}^{E},\ldots ,P_{mpq}{}^{E},\ldots ,P_{11\;40\;12}{}^{E} \end{array} \end{array}\right]$$
$$\left(\begin{array}{ccc} m=1,2,\ldots ,11 & p=1,2,\ldots ,40 & q=1,2,\ldots ,12 \end{array}\right)$$

Finally, $P^{I}$ and $P^{E}$ are taken into Equation (8) to calculate the relative entropy.
(11)$$H_{R1}\left(X\right)=\underset{x\in X}{\sum }P^{I}{}_{1}\left(x\right)log\frac{P^{I}{}_{1}\left(x\right)}{P^{E}{}_{1}\left(x\right)}$$

For the convenience of comparison, the relative entropy of each term of SED magnitude, circular distribution and radial distribution are calculated separately. The evaluation results of relative entropy of SED distribution in four error cases are shown in Table 3.

Comparing with the results in Figure 12 and Table 3, the following conclusions can be obtained:In error case 1, the relative entropy of circular distribution and radial distribution are small in four cases, but its relative entropy of SED magnitude is large. It means that the SED magnitude distribution of error case 1 is very different from that in the ideal model even though its SED position distribution is the most similar to the ideal model.Case 2 and case 3 are in the middle of the calculation results. Although their relative entropy in the radius direction are larger, their overall relative entropy are also in the middle.In case 4, the overall relative entropy is the smallest, and the relative entropy of SED magnitude is the smallest too. Its relative entropy of the circular and radial distribution is only greater than that in error case 1. This means that the overall SED distribution of case 4 is the most similar with that of the ideal case in Figure 8a, even though the SED circular and radius distribution of case 4 is not as good as that of case 1.

## 3. Simulation and Optimization

### 3.1. Optimization of Assembly Process

The magnitude and distribution of stress are important performance indexes of the lens. However, in traditional tolerance analysis, a revolving body is regarded as a part with macro tolerance. Its assembly angle does not have any impact on the accuracy of the optical lens system, and there is no need to optimize it. Taking the typical optical lens system shown in Figure 2 as an example, the spacer assembly angle *θ* is defined as the difference between initial angle of spacer and the actual assembly angle (taking the ideal axis of the lens barrel as the axis), as shown in Figure 13.

In the assembly process, due to the existence of geometric errors, the assembly angle *θ* will have a great impact on the final assembly accuracy and the magnitude and position distribution of part stress. In order to solve this problem, this paper proposes an optimization method of precision assembly process. In this method, the assembly angle *θ* of the revolving part is taken as the optimization variable; the relative entropy $H_{r}$ of the part’s SED distribution minimum is taken as the objective function; the constraint condition is that the maximum stress $\sigma_{max}$ of the part is less than the average value of the maximum stress of all models:(12)$$\begin{array}{c} \theta \in \left[0,2\pi \right]\\ \underset{\theta }{\min}H_{r}\left(\theta \right)\\ s.t.\;\;\;\sigma_{max}\left(\theta \right)\leq \bar{\sigma_{max}}\left(\theta \right) \end{array}$$

The optimization process is shown in Figure 14. Firstly, the measurement and error modeling of the mating surfaces of the parts are carried out, and the three-dimensional model with real errors is established [2]. Then, the error model is imported into the finite element software. The assembly angle of parts is changed for multiple calculations. The position coordinates, SED of elements and the maximum stress of the model are extracted from the calculation results. The optimal result is the assembly angle whose relative entropy is minimum and maximum stress is less than the average.

### 3.2. Simulation Setup

In this simulation, the front parts of an optical lens system are intercepted, including the lens barrel, the lens, the spacer and the threaded ring. As shown in Figure 15, all parts in the model are revolving body. The spacer and the lens are successively assembled in the lens barrel, fixed by the threaded ring with preload *F*. The material of the threaded ring, the spacer and the lens barrel is 7075 aluminum alloy, and the lens material is optical glass. The assembly angle of the spacer is *θ*. In the finite element software, the spacer can be directly rotated by setting the rotation axis and angle. The preload and assembly angle of the threaded ring remain unchanged in the whole optimization process.

The bottom surface of the threaded ring is defined as error surface A, and the top surface of the spacer is defined as error surface B. The spacer and the threaded ring are imported according to the measuring coordinate system of error surface A and B, and the angle is set as the initial assembly angle of the part, that is, *θ* = 0. The error surface A and B are matched and the model is calculated multiple times with changing of the assembly angle *θ* of the spacer. The assembly angle *θ* corresponding to the model with the minimum relative entropy and satisfying the stress constraints in all finite element calculation results is taken as the optimization result.

Firstly, coordinate measuring machines (CMM) are used to measure the morphology of the contact surfaces between the threaded ring and the lens (error surface A), the spacer and the lens (error surface B), and the coordinate point cloud file is obtained. Then, geometric error modeling [2] is carried out to obtain the models with error surfaces of the spacer and the threaded ring. The error surfaces A’s and B’s coordinates are measured by Edwardian Daisy CMM whose measurement uncertainty is 1.7 μm. Since the total standard deviation of the measurement data is between 10 μm and 20 μm, the measurement uncertainty can be ignored. The error surface A of the threaded ring is measured four circles, with 60 points per circle; the error surface B of the spacer is measured two circles, with 60 points per circle. In the CAD software, the models of threaded ring and spacer with geometric error are established, as shown in Figure 16.

The models of the spacer and the threaded ring with geometric errors are assembled with an ideal lens and lens barrel, and then imported into the finite element software for mesh generation, boundary condition setting and load application. The finite element parameters are shown in Table 4. The explicit solver is chosen. Since, in the implicit solver, each model needs to be adjusted separately to achieve convergence, and the parameter settings might be slightly different, which leads to the decrease of the comparability of the calculation results. The elements number in a mechanical contact problem is vital for the results. Some studies performed sensitivity analysis to adjust their finite element models [28,29]. The sensitivity analysis of elements number is carried out and the result is shown in Figure 17a. Considering the stability of the results, 83,136 elements are chosen. Only the elasticity of materials is considered, not plasticity. Since the maximum stress in the results is not more than 50 MPa, which is far lower than the plasticity of 7075 aluminum alloy (about 455 MPa). The maximum stress of the lens is not more than 11 MPa, which is far lower than the breaking point of glass H-ZK10. After the investigation in the early stage, the empirical preload of the threaded ring is 150 ± 50 N. Therefore, the preload is set to 150 N, which remains unchanged in the whole optimization process. The error surface is measured by 60 measuring points per circle, that is, measured every 6°. Considering the measuring points distance and computer calculation ability, 20° is selected as the minimum variation of assembly angle of the spacer. In the finite element software, the ideal axis of the lens barrel in Figure 13 is taken as the rotation axis, and the spacer is rotated every 20° to obtain a total of 18 error models with different angles from 0° to 340°. At the same time, an ideal model without error surfaces is established in the same way. All the finite element models are calculated, and the SED, element coordinates, and element stress of the lens are obtained.

In order to confirm whether the result of the explicit solver is quasi-static, and whether the setting of the integration time is appropriate, we derive the kinetic energy history, and the result is as shown in the Figure 17b. The load is divided into 6 steps and gradually loaded. It can be seen that, except for the beginning, the whole curve is relatively gentle, which means that the integration time is suitable and the simulation results are credible.

### 3.3. The Relative Entropy Evaluation of SED Distribution

The SED nephograms of the error models are shown in Figure 18. For the convenience of comparison, the scales of the cloud charts are unified, and the maximum stress $\sigma_{max}$ is marked in each chart.

Firstly, *M*, *P*, and *Q* are selected according to the method in Section 2.3.1. The error model with *θ* = 0 is selected for calculation. With the increasing of *M*, *P* is kept the initial value (*P* = 200) unchanged, and *Q* is increased from 1 to 5 successively for calculation. The results are shown in Figure 19a. *M* = 13 is taken at the place where the curve goes smoothly. *M* = 13 is kept unchanged, *P* is increased, and then *Q* is increased from 1 to 3 for calculation. The results are shown in Figure 19b. *P* = 700 is taken at the place where the curve is smooth. Keeping *M* = 13 and *P* = 700, the *Q* value is increased in turn for calculation, and the result is shown in Figure 19c. *P* = 700 is taken at the place where the curve is smooth.

So far, the values of *M*, *P*, and *Q* are determined:M=13 P=700 Q=50

The relative entropy of SED distribution is calculated using the method in Section 2.2. The elements with SED greater than median are selected as SED concentration elements. According to Equation (4), the number ($N_{lc}$) of SED concentration elements in each region divided by *M*, *P*, and *Q* is calculated in turn. According to Equation (6) and Equation (7), the SED probability distributions of ideal model $P^{I}$ and error model $P^{E}$ are constructed. The $P^{I}$ and $P^{E}$ are taken into Equation (8), and the relative entropy is calculated. The results of 18 different angle error models are shown in Figure 20.

The relative entropy results in Figure 20 show a regular pattern with the changing of assembly angle. At 160° and 340° the relative entropy reaches the lowest point, and their angle difference is 180°. At 60° and 240° the relative entropy reaches the highest point, and their angles difference is also 180°. The morphology of error surfaces A and B may relate to this. In Figure 16, it can be clearly seen that the spacer and the threaded ring presents a “saddle” type. With the rotation of the spacer, the matching of the high point and low point of the error surface A and B changes, resulting in the regular change of the SED distribution of the lens.

### 3.4. Results and Discussion

The maximum stress of the lens is a very important index, so it is considered as the constraint for optimization. The relative entropy of lens SED distribution has been calculated in Section 3.3. The maximum lens stress of 18 error models is used to calculate the average value, as shown in Figure 21. The relative entropy of the model in which the stress is greater than the average value is marked as grey.

Figure 21 shows that the assembly angle *θ* with the minimum relative entropy of the SED distribution is 160°, which means that the SED distribution on the lens is the closest to the ideal model, but its maximum stress cannot meet the constraints. Among the error models satisfying the constraint $\sigma_{max}\left(\theta \right)\leq \bar{\sigma_{max}}\left(\theta \right)$, the error model of 140° has minimum relative entropy. That is to say, the optimization result is:$$\theta_{opt}=140{}^{\circ}$$

The SED nephograms of Figure 18 can also show the optimization results. In the case of uniform scale, only the error models of 140°, 160°, 340° keep the lens’ center SED at a small level (blue area). The relative entropy of these three error models are also low. But the upper stress limits are very high in the 160° and 340° SED nephograms. At the same time, other error models with low upper stress limits have larger center SED (green area) distributed in the lens, comparing with the error models of 140°, 160°, and 340°. Hence the final optimization result is $\theta_{opt}=140{}^{\circ}$.

The advantage of this method is the overall evaluation consider the magnitude and the position. It avoids the one sidedness of the optimization method which only considers the maximum stress or the uniformity of stress (or SED) magnitude without considering the stress (SED) position distribution. The method is intuitive, practical and operable.

## 4. Conclusions

In this paper, the relative entropy evaluation method of stress distribution characterized by strain energy density (SED) distribution is proposed. The relative entropy is used to characterize the difference of stress distribution between the error model and the ideal model. Minimum relative entropy means that the stress distribution of the error model approaches the ideal model to the greatest extent. The validity of the method is verified by a case.Taking the relative entropy of SED distribution as the objective and the maximum stress lower than the average value of all models as the constraint condition, the assembly angle of the spacer in the optical lens system is optimized. After optimization, the stress distribution of the lens is closer to that of the ideal model, and the maximum value of stress is lower than the average value of it.The relative entropy evaluation method of stress (SED) distribution and optimization method proposed in this paper are suitable for single assembly parameter optimization (assembly angle) of a revolving body. The research on stress (SED) distribution evaluation and multi-assembly parameter optimization of arbitrary shape entities will be carried out in the future.

## Figures and Tables

**Figure 1 entropy-22-00137-f001:**
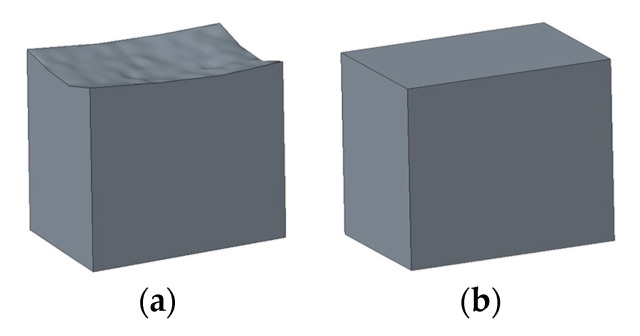
(**a**) Cube model with error surfaces. (**b**) Cube model with ideal surfaces.

**Figure 2 entropy-22-00137-f002:**
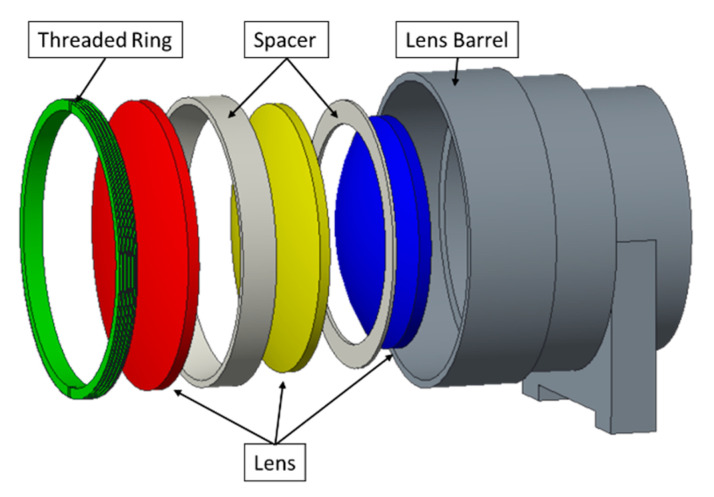
Optical lens system structure.

**Figure 3 entropy-22-00137-f003:**
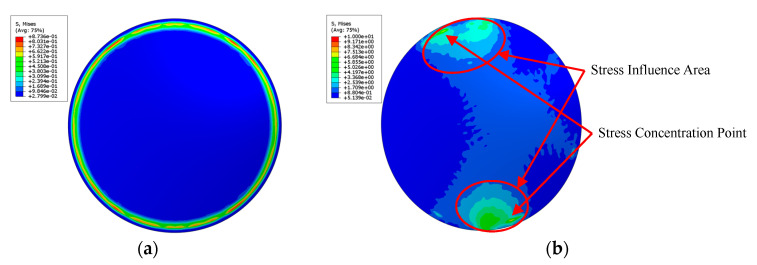
(**a**) The ideal spacer contacts the lens uniformly, and the stress is evenly distributed in the circular direction. (**b**) The spacer with geometric error causes the stress concentration of the lens.

**Figure 4 entropy-22-00137-f004:**
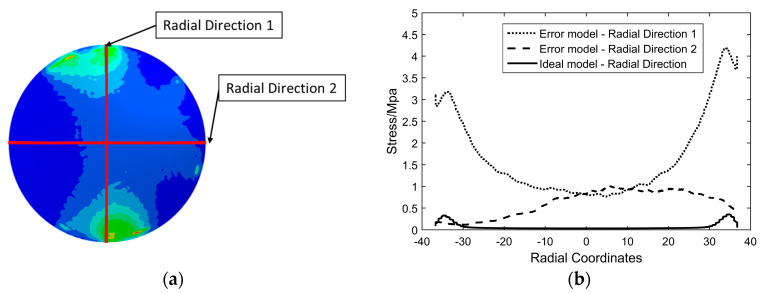
(**a**) Stress nephogram of an error model. (**b**) Radial stress curves of error model and ideal model.

**Figure 5 entropy-22-00137-f005:**
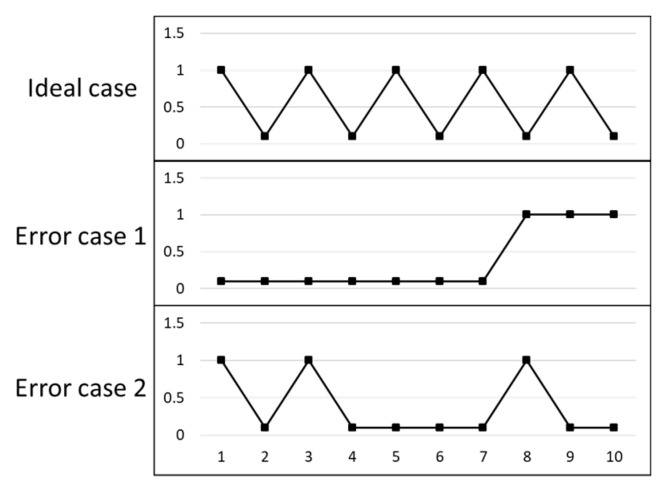
A hypothetical ideal case and two error cases.

**Figure 6 entropy-22-00137-f006:**
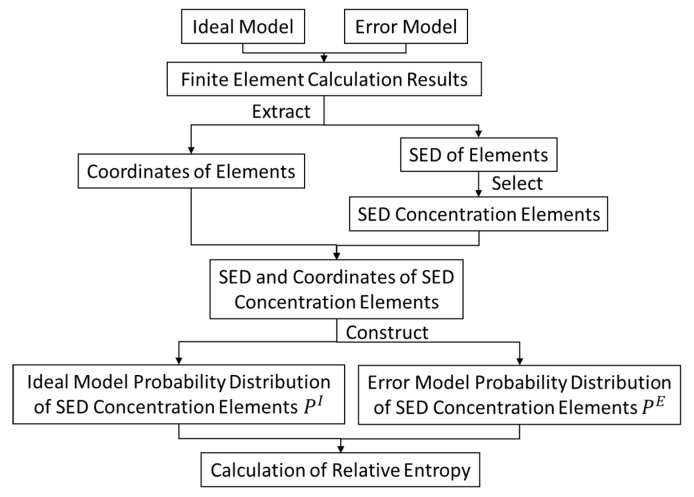
Process of relative entropy evaluation method of the SED distribution.

**Figure 7 entropy-22-00137-f007:**
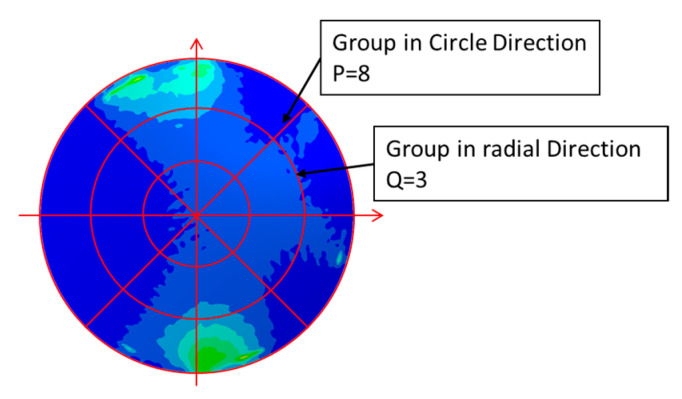
The schematic diagram of lens region division (*P* = 8, *Q* = 3).

**Figure 8 entropy-22-00137-f008:**
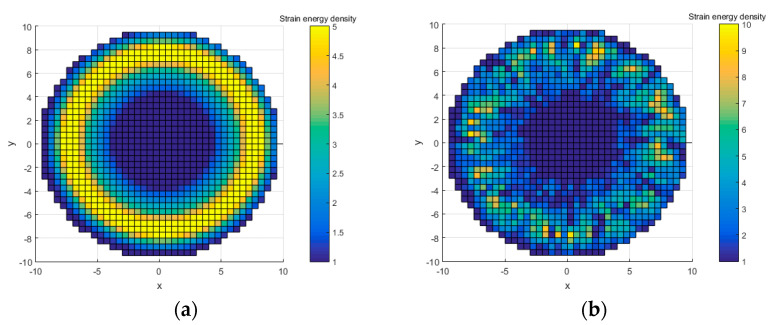
(**a**) An ideal case settled according to the stress change pattern of ideal model in Figure 4. (**b**) A randomly generated error model: error case 1.

**Figure 9 entropy-22-00137-f009:**
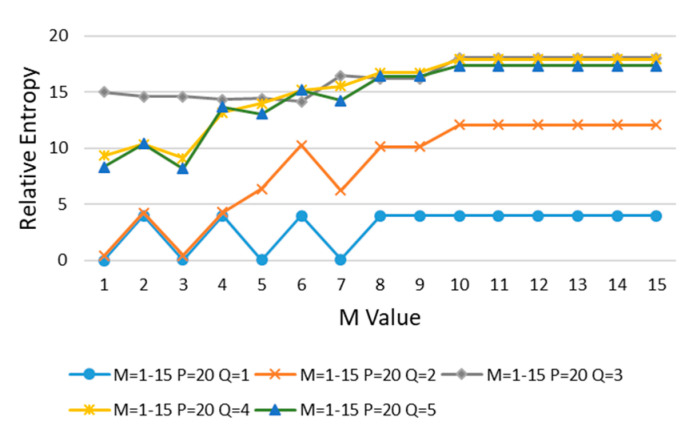
*M* is increased, *P* = 20, and *Q* is increased from 1 to 5.

**Figure 10 entropy-22-00137-f010:**
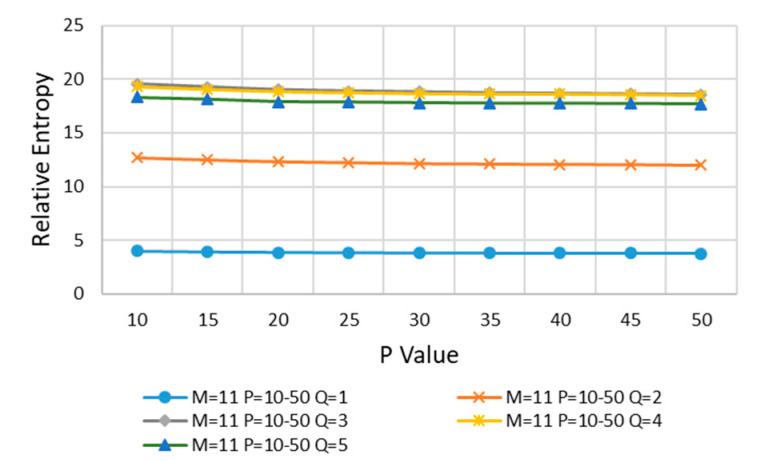
*M* = 11, *P* is increased, and then *Q* is increased from 1 to 5.

**Figure 11 entropy-22-00137-f011:**
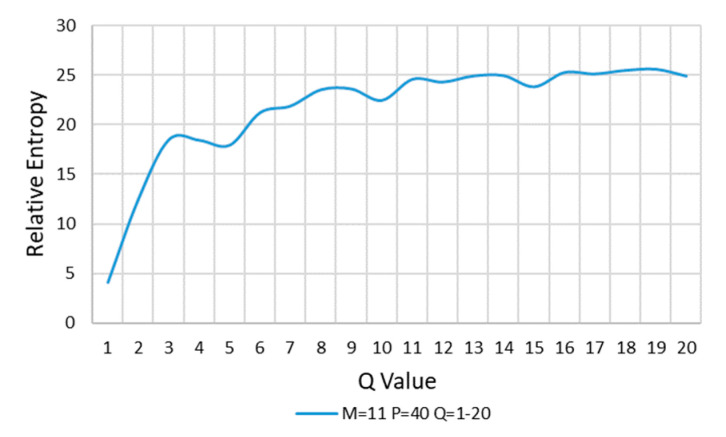
*M* = 11, *P* = 40, and *Q* is increased.

**Figure 12 entropy-22-00137-f012:**
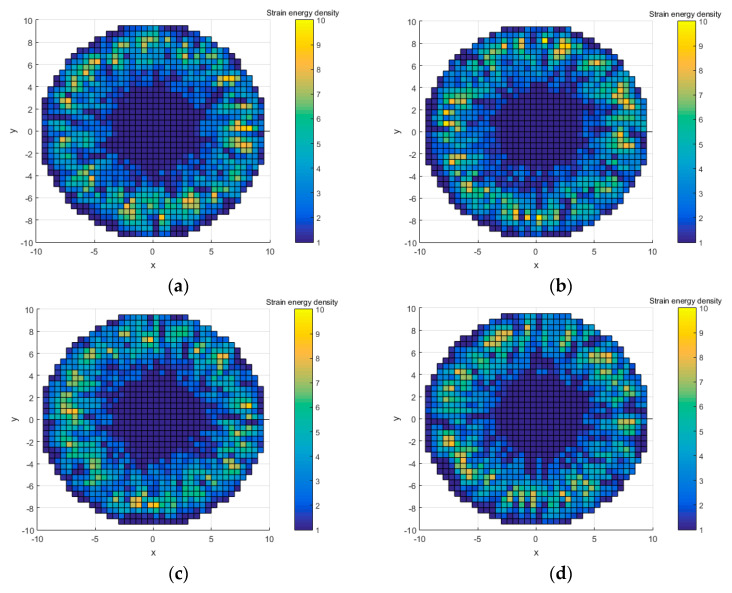
Four randomly generated error models. (**a**) Error case 1; (**b**) Error case 2; (**c**) Error case 3; and (**d**) Error case 4.

**Figure 13 entropy-22-00137-f013:**
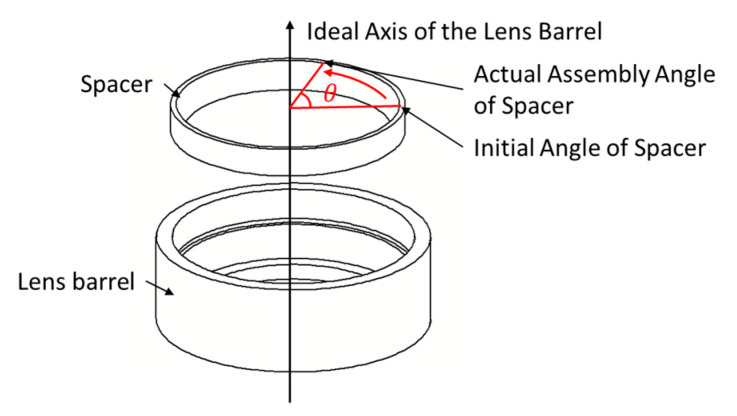
Schematic diagram of the spacer assembly angle.

**Figure 14 entropy-22-00137-f014:**
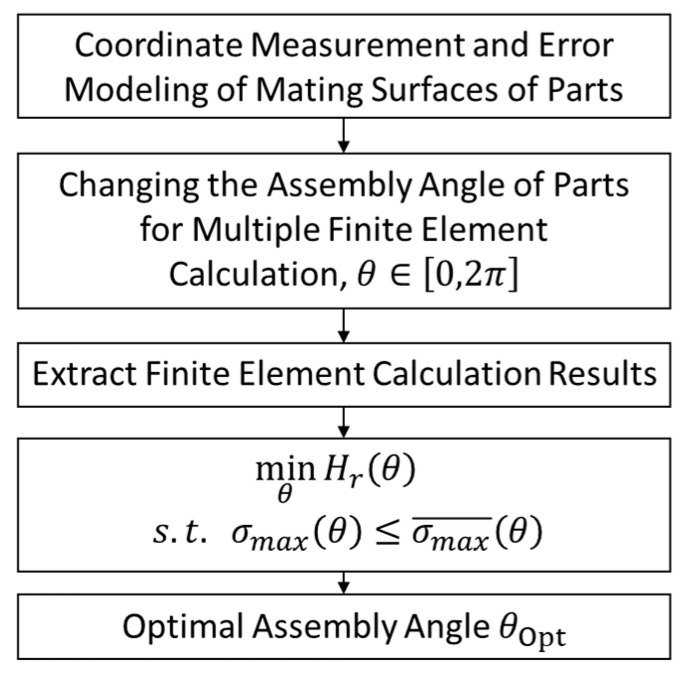
Assembly angle optimization process.

**Figure 15 entropy-22-00137-f015:**
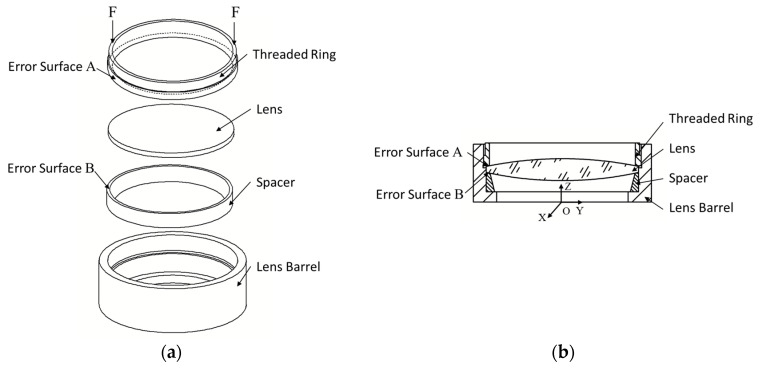
Optical lens system model. (**a**) Exploded view. (**b**) Sectional view.

**Figure 16 entropy-22-00137-f016:**
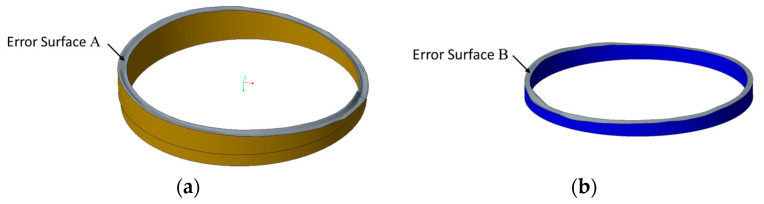
Models with error surfaces (150× magnification). (**a**) Error surface A of the threaded ring. (**b**) Error surface B of the spacer.

**Figure 17 entropy-22-00137-f017:**
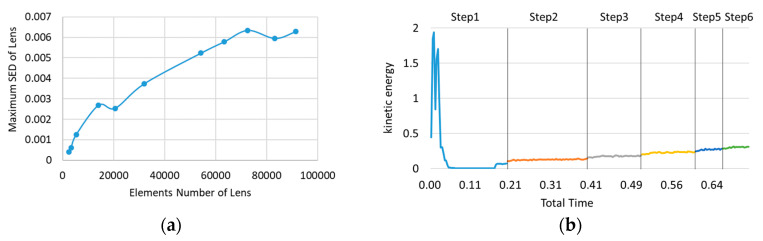
(**a**) The sensitivity analysis of elements number. (**b**) Kinetic energy history of the result.

**Figure 18 entropy-22-00137-f018:**
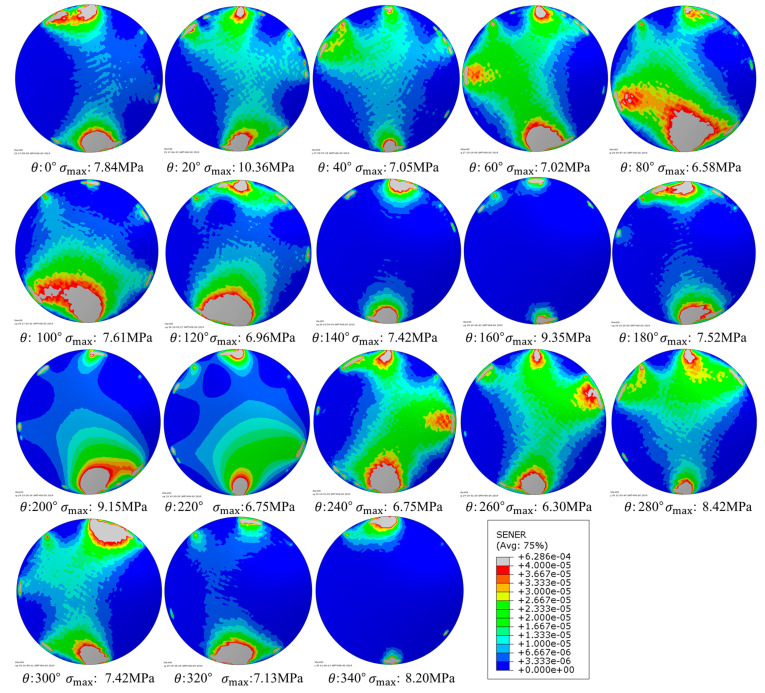
SED nephograms of the error model from 0° to 340°.

**Figure 19 entropy-22-00137-f019:**
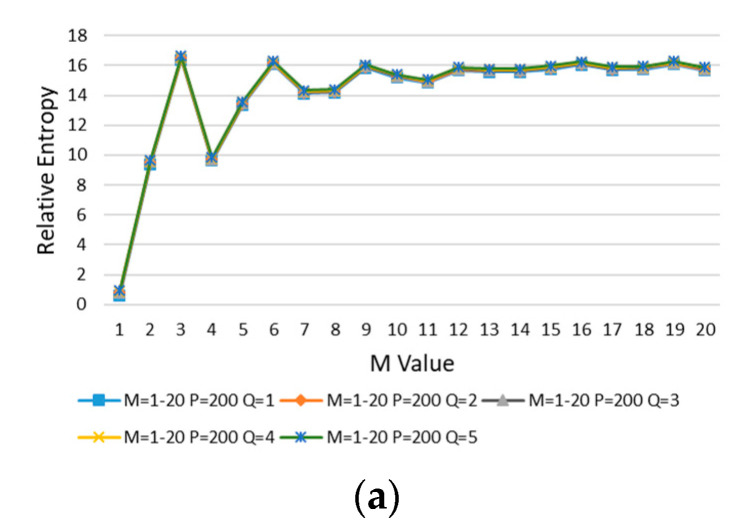
(**a**) *M* is increased, *P* = 200, *Q* from 1 to 5. (**b**) *M* = 13, *P* is increased, *Q* from 1 to 3. (**c**) *M* = 13, *P* = 700, *Q* is increased.

**Figure 20 entropy-22-00137-f020:**
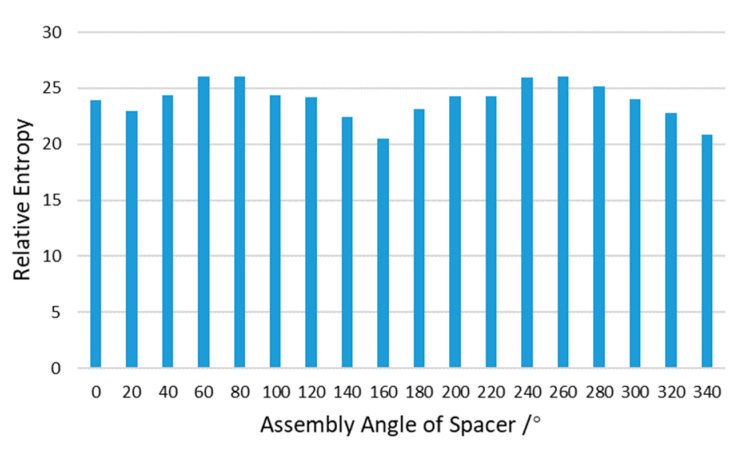
Relative entropy evaluation results of SED distribution.

**Figure 21 entropy-22-00137-f021:**
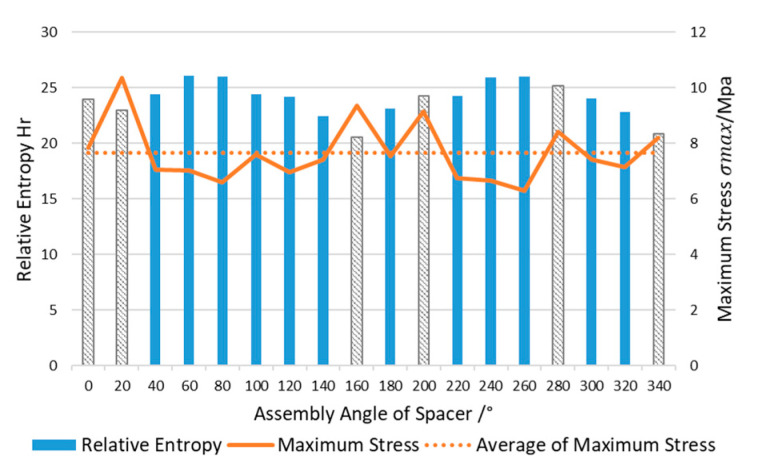
Relative entropy evaluation of SED distribution and maximum stress of the lens.

**Table 1 entropy-22-00137-t001:** The parts and functions of optical lens system.

Parts	Functions
Lens Barrel	Provides support and installation for internal parts
Lens	Optical element
Spacer	Separates the lens and keeps it at a proper distance
Threaded ring	Provides preload to fix lens and spacer

**Table 2 entropy-22-00137-t002:** The entropy and relative entropy calculation results of the two error cases in Figure 5.

Case	Entropy	Relative Entropy
Error case 1	2.516	2.008
Error case 2	2.516	1.617

**Table 3 entropy-22-00137-t003:** Relative entropy evaluation results.

Evaluation Category	Relative Entropy of SED Magnitude	Relative Entropy of Circular Distribution	Relative Entropy of Radial Distribution	Overall Relative Entropy
Divided Regions	*M* = 12 *P* = 1 *Q* = 1	*M* = 1 *P* = 40 *Q* = 1	*M* = 1 *P* = 1 *Q* = 11	*M* = 12 *P* = 40 *Q* = 11
Error Case 1	27.5785	0.0063	4.2619	25.2012
Error Case 2	26.1303	0.0071	4.5927	24.5225
Error Case 3	24.4327	0.0075	4.5273	24.0603
Error Case 4	24.0100	0.0071	4.3931	23.0929

**Table 4 entropy-22-00137-t004:** Parameters of the finite element model.

Terms	Parameters	Values
Software setting	Software	ABAQUS
	Type of contact	Surface-to-surface
	Type of simulation	Explicit
Elements	Type of elements	C3D8R
	Formulation of elements	Linear
	Elements Number of Lens	83,136
Material	H-ZK10 (Lens)	Young’s modulus: 82.51 GPa Poisson’s ratio: 0.27
	7075 aluminum alloy (Lens barrel, threaded ring, spacer)	Young’s modulus: 70 GPa Poisson’s ratio: 0.32
Boundary conditions	Full restraint at the bottom of the barrel	ENCASTRE
Load	Preload of threaded ring	150 N (Empirical value)

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
