# Peer review of "An Optimization Method of Precision Assembly Process Based on the Relative Entropy Evaluation of the Stress Distribution"

_entropy, 2020, doi:10.3390/e22020137_

Round 1

Reviewer 1 Report

Thank you very much for inviting me to review this manuscript entitled “An Optimization method of precision assembly process based on relative entropy evaluation of stress distribution”.

Major corrections:

The manuscript is poorly focused as a methodology is proposed, and as a practical case it is applied to optical lens. Authors should explain this point in the abstract and in the introduction as it is not clear.

The introduction must be expanded and improved since it is very poor. Nothing is described about the method that the authors propose and based on what references it is made.

The manuscript is also very poorly structured. Why have the authors explained the results of Von Mises tensions in a lens and then explain an entropy problem? Please correct this error since the manuscript is gibberish in the form in which it has been submitted.

Regarding the proposed Finite Element Model. Authors must specify the following:

What is the reason for using von Mises stress in the study? Why have the authors not used the principal streses, contact streses or any other type of stress? Wouldn't it be more useful to use deformation energy or simply deformation? Please specify What is the reason for choosing an analysis with an explicit solver instead of an analysis with an implicit solver? What has been the integration time chosen in the analysis and how have they determined it (this point is very important)? The authors should show this type of study, in addition to the kinetic energy required in the analysis in order to determine the influence of solver explicit on the result. What are the properties of the materials used? Has plasticity been considered? Please specify. The authors should know that the size of the elements in a mechanical contact problem is vital for the determination of the stresses in the contact area and around. The authors have conducted a sensitivity analysis to determine what has been the appropriate size of the elements used in their simulations? What has been the size of the item used? The following papers show how their authors have performed a sensitivity analysis to adjust their finite element models (I suggest you consult them as they are FEM models that have been adjusted for mechanical problems).

Íñiguez-Macedo, S., Lostado-Lorza, R., Escribano-García, R., & Martínez-Calvo, M. Á. (2019). Finite Element Model Updating Combined with Multi-Response Optimization for Hyper-Elastic Materials Characterization. Materials, 12(7), 1019.

Gómez, F., Lorza, R., Bobadilla, M., & García, R. (2017). Improving the Process of Adjusting the Parameters of Finite Element Models of Healthy Human Intervertebral Discs by the Multi-Response Surface Method. Materials, 10(10), 1116.

Reviewer 2 Report

In this paper the authors propose the use of relative entropy to compare the strain energy density (SED) distribution difference between ideal and error assembly models to evaluate and optimize assembly results. The authors describe the rationale behind their approach, which can be reduced to the fact that the relative entropy take into account with a single number of both magnitude and spatial distribution of the field of interest, propose an example to describe their solution protocol, and then apply their method to the assembly of a spacer and lens system. Overall the manuscript if written with sufficient clarity and abundance of data, and the results are interesting. However the manuscript is also too long compared to the content, and more importantly it does not make clear enough what the real message is, whether the method or the chosen application. Given the journal to which the manuscript is submitted and that the authors claim that their method is "universal" (line 124), we would think that the proposed methodology and the role of entropy in it has to have the main focus, whereas the current structure of the manuscript does not reflect that.

Our main concerns are:

1) The introduction is confusing and should be restructured and rewritten. In present form, the flow is unclear, as after a too brief paragraph the CAD of the application case (figure 1) are immediately introduced with too little background, giving the impression that the application is what the authors care about mostly. Then Figure 2 and 3 are already presenting stress distributions, without mentioning where these come from nor the software/method used to compute them. And only then background on entropy methods with references are given. And figure 4 comes too early, as it is only introduced as an example of the method.
Briefly, the introduction should be restructured and presented with a more conducing flow, giving first the background on entropy methods, and why one of them has been chosen here, and what are prior related works, and how the method works, and finally what application has been chosen by the authors to demonstrate its worth compared to alternatives.
Moreover, there are several instances where the authors claim what the paper is about, whereas we get the impression that the most important is what stated in lines 81-82. If that is the case, the paper should be presented in a way consistent with that aim.

2) Some if not many of the data presented are not necessary for the understanding of the method, and could be put in an appendix given that the paper is quite long and redundant, for instance, figures 17-19.

3) The paper does not give a way to compare the result of the assembly optimization method proposed to the result of any alternative optimization method. Such comparison is required for fairness and to make the case stronger.

Minor details (in order of appearance):

a) Line 18: space before "and"

b) Line 32-5: totally unclear the relevance of this mention at this point of the paper, this should be put in the application part of the manuscript, after the intro and background.

c) Figure 2 and 3: where do these come from? how was the data acquired?
There is also an error in caption of figure 3b (modle --> model)

d) We suggest to put the discussion around figure 4 all in one place instead in both Introduction and Methodology as in current form.

e) The formatting of the figures is not uniform, and the quality of the plots in particular should be improved.

f) Equation 11 is exactly the same as equation 8.

Round 2

Reviewer 1 Report

Dear Authors:

All questions and suggestions proposed by the reviewer have been correctly answered and therefore, from my point of view, the paper can be published in Entropy Journal

¡¡¡ Congratulations for your work ¡¡¡

Reviewer 2 Report

The authors answered precisely to our queries and consequently revised significantly their original submission. We think the paper is now more convincing and better presented, in a form that can now be recommended for publication.